# FEINT IN MULTI-PLAYER GAMES

## ABSTRACT

This paper introduces the first formalization, implementation and quantitative evaluation of *Feint* in Multi-Player Games. Our work first formalizes *Feint* from the perspective of Multi-Player Games, in terms of the temporal, spatial and their collective impacts. The formalization is built upon *Non-transitive Active Markov Game Model*, where *Feint* can have a considerable amount of impacts. Then, our work considers practical implementation details of *Feint* in Multi-Player Games, under the state-of-the-art progress of multi-agent modeling to date (namely Multi-Agent Reinforcement Learning). Finally, our work quantitatively examines the effectiveness of our design, and the results show that our design of Feint can (1) greatly improve the reward gains from the game; (2) significantly improve the diversity of Multi-Player Games; and (3) only incur negligible overheads in terms of time consumption. We conclude that our design of Feint is effective and practical, to make Multi-Player Games more interesting.

## 1 INTRODUCTION

Game simulations, which only use Markov Game Model (Filar (1976)) or its variants (Wampler et al. (2010); Kim et al. (2022)), breed the needs for the diversity and the randomness to improve the game experiences. The trends of evolving more details into simulated games demand: ❶ the need for non-transitivity (i.e. there are no dominant gaming strategies), which allow players to dynamically change game strategies. In this way, the newly-incorporated strategies can maintain a high level of the diversity, which guarantee a high extent of unexploitability (Liu et al. (2021)); and ❷ the strict requirements on temporal impacts (and its implications on spatial and collective impacts), since modern game simulations are highly time-sensitive ( Nota & Thomas (2020)). Therefore, new optimizations on these game models are expected to be elegant and easy-to-implement, to preserve the original spirits of these games.

Our work first builds upon representative examples from the above two trends, by unifying two state-of-the-art progress of Multi-Player Games: ❶ we use Unified Behavioral and Response Diversity (described in Liu et al. (2021)), which exploits non-transitivity (i.e. no single dominant strategy in many complex games), to highlight the importance of the diversity in game policies. Moreover, we address the issue from their work, which fails to consider the intensity and future impacts from complex interactions among agents; and ❷ we incorporate Long-Term Behavior Learning (described in Kim et al. (2022)), which proposes Active Markov Game Model to emphasize the convoluted future impacts from complex interactions among agents. Based on the above two results, we unify them as a new model called *Non-transitive Active Markov Game Model* (NTAMGM), and use it throughout this work. This unification satisfies the need for a game model where (A) agents have intense and time-critical interactions; and (B) the design space of game policies is highly diverse. The definition of *NTAMGM* is described below.

● **Non-transitive Active Markov Game Model**: We define a $K$-agent *Non-transitive Active Markov Game Model* as a tuple $\langle K, S, A, P, R, \Theta, U \rangle$: $K = \{1, ..., k\}$ is the set of $k$ agents; $S$ is the state space; $A = \{A_i\}_{i=1}^{K}$ is the set of action space for each agent, where there are no dominant actions; $P$ performs state transitions of current state by agents' actions: $P : S \times A_1 \times A_2 \times ... \times A_K \to P(S)$, where $P(S)$ denotes the set of probability distribution over state space $S$; $R = \{R_i\}_{i=1}^{K}$ is the set of reward functions for each agent; $\Theta = \{\Theta_i\}_{i=1}^{K}$ is the set of policy parameters for each agent; and $U = \{U_i\}_{i=1}^{K}$ is the set of policy update functions for each agent.

Based on the above assumption of Multi-Player Games, our goal is to **incorporate Feint, a set of actions to mislead opponents, for strategic advantages in Multi-Player Games**. Prior works simply incorporate Feint in the context of Two-Player Games (e.g. Wampler et al. (2010); Won et al. (2021a)), and our works begins by addressing the limitations of the derived version (denoted as the basic formalization of Feint) from these works. We find that: the basic formalization of Feint overlooks the complexity of potential impacts in Multi-Player Games, and therefore can not be generalized for Multi-Player Games. To this end, we deliver the first comprehensive formalization of Feint, by separating the complex impacts into ❶ the temporal dimension; ❷ the spatial dimension; and ❸ the collective impacts from these two dimensions. We also show that how the above components of our formalization can be synergistically put together. Based on the proposed formalization, we clear the implementation roadmap, under both Inference Learning and Reinforcement Learning models, to justify the applicability of our proposed formalization.

To properly examine the benefits of our method, we first extensively build two complex scenarios, using Multi-Agent Deep Deterministic Policy Gradient (MADDPG Lowe et al. (2017)) and Multi-Agent Actor-attention Critic (MAAC Iqbal & Sha (2019)), with six agents in total. Then, we implement our formalization upon these two extensively-engineered scenarios. Our quantitative evaluations show that our formalization and implementations have great potential in practice. We first show that our work can make the game more interesting, via the following two metrics: for the Diversity Gain, our method can increase the exploitation of the search space by 1.98X, measured by the Exploitability metric; and for Gaming Reward Gain, our method can achieve 1.90X and 2.86X gains, when using MADDPG and MAAC respectively. We then show that our method only incur negligible overheads, by using per-episode execution time as the metric: our method only introduces less than 5% more for the time consumption. We conclude that our design of Feint is effective and practical, to make Multi-Player Games more interesting.

## 2 BACKGROUND AND MOTIVATION

### 2.1 EXISTING MARL MODELS

Multi-Agent Reinforcement Learning (MARL) aims to learn optimal policies for agents in a multi-agent environment, which consists of various agent-agent and agent-environment interactions. Many single-agent Reinforcement Learning methods (e.g. DDPG Lillicrap et al. (2016), SAC Haarnoja et al. (2018), PPO Schulman et al. (2017) and TD3 Fujimoto et al. (2018)) can not be directly used in multi-agent scenarios, since the rapidly-changing multi-agent environment can cause highly unstable learning results (evidenced by Lowe et al. (2017)). Thus, recent efforts on MARL model designs aim to address such an issue. Foerster et al. (2018) proposes Counterfactual Multi-Agent (COMA) policy gradients, which uses centralised critic to estimate the Q-function and decentralised actors to optimize agents' policies. Lowe et al. (2017) proposes Multi-Agent Deep Deterministic Policy Gradient (MADDPG), which decreases the variance in policy gradient and instability of Q-function of DDPG in multi-agent scenarios. Iqbal & Sha (2019) proposes Multi-Agent Actor-attention Critic (MAAC), which applies attention entropy mechanism to enable effective and scalable policy learning. These models can have varied impacts within a diverse set of scenarios.

### 2.2 FEINT IN A NUTSHELL

*Feint* is common for human players, as a set of active actions to obtain strategic advantages in real-world games. Examples can include sports games such as boxing, basketball and car racing Güldenpenning et al. (2017; 2018); Hyman (1989), and electronic games such as King of Fighters and Starcraft Team (2021); Critch & Churchill (2021). Though *Feint* is undoubtedly important in game simulations, there still lacks a comprehensive formalization of *Feint* for Non-Player Characters (NPCs) in Multi-Player Games. Only a limited amount of works tackle this issue. Wampler et al. (2010) is an early example to incorporate *Feint* as a proof-of-concept, which focuses on constructing animations for nuanced game strategies for more unpredictability from NPCs. More recently, Won et al. (2021a) uses a set of pre-defined *Feint* actions for the animation, which further serves under an optimized version of control strategy based on Online Reinforcement Learning (i.e. in animating combat scenes). However, these prior works (1) solely focus on Two-Player Games, which can not be effectively generalized to multi-player scenarios; and (2) lack an comprehensive exploration of potential implications from *Feint* actions in game strategies.

## 2.3 NOVELTY OF OUR WORK

The novelty of our work is three-folded. First, our work introduces the first formalization of *Feint*, which can be generalized to Multi-Player Games. Prior works solely focuses on Two-Player Games, which have the flexibility and scalability issue from the basic formalization. Second, our work provides effective implementations of *Feint* in Multi-Player Games, by exploiting our formalization appropriately on common parts of MARL models (i.e. the reward function). Our formalization can be applied to existing MARL models, and is expected to be applicable in future MARL models. Third, our work identifies the unique characteristics of *Feint*, by differentiating *Feint* with other regular actions. Hence, our work is expected to be applicable in different scenarios, with only a limited amount of refinements.

## 3 FEINT FORMALIZATION

### 3.1 THE BASIC FORMALIZATION: DERIVATION AND LIMITATIONS

We summarize two major limitations of existing works to justify that they cannot deliver a sufficient formalization of *Feint* in Multi-Player Games. Since there are no prior formalization, we discuss relevant works and derive the key features to discuss them in detail.

❶ The basic formalization on temporal impacts is insufficient for Multi-Player Games. Multi-Player Games require agents to account for future planning for decision-making, which is critical for deceptive actions like *Feint* Mnih et al. (2013); Naik et al. (2019); Nota & Thomas (2020). Several works simplify the temporal impacts of deceptive game strategies in different gaming scenarios. Mnih et al. (2013) uses a discount factor $\gamma$ to calculate the reward for following actions as $\sum_{t=0}^{\infty} \gamma^t R^i(s_t, a_t^i, a_t^{-i})$ for agent $i$. However, such a method suffers from the "short-sight" issue Naik et al. (2019), since the weights for future actions' rewards shrink exponentially with time, which are not suitable for all gaming situations (discussed in Nota & Thomas (2020)). More recently, Kim et al. (2022) applies a long-term average reward, to equalize the rewards of all future actions as $\frac{1}{T} \sum_{t=0}^{T} R^i(s_t, a_t^i, a_t^{-i})$ (i.e. for agent $i$). However, such a method is restricted by the "far-sight" issue, since there are no differentiation between near-future and far-future planning. The mismatch between abstraction granularity heavily saddles with the design of *Feint*, because they use relatively static representations (e.g. static $\gamma$ and $T$). Therefore, they cannot be aware of any potential changes of strategies in different phases of a game. Hence, the temporal dimension is simplified for the basic *Feint* formalization.

❷ The basic formalization cannot be effectively generalized to Multi-Player Game scenarios. Prior works, which attempt to fuse *Feint* into complete game scenarios, only consider two-player scenarios Won et al. (2021a); So et al. (2022). However, in Multi-Player (more then two player) Games, gaming strategies (especially deceptive strategies) yield spatial impacts on other agents. Such impacts have been overlooked by all prior works. This is because an agent, who launches the *Feint* actions, can impact not only the target agent but also other agents in the scenario. Therefore, the influences of such an action needs to account for spatial impacts Liu et al. (2021). Moreover, with a new dimension accounted, the interactions between these two dimensions also raise a potential issue for their mutual collective impacts.

### 3.2 OUR FORMALIZATION: GENERALIZED FOR MULTI-PLAYER GAMES

Therefore, to deliver an effective formalization of *Feint* in Multi-Player Games, it's essential to consider the temporal, spatial and their collective impacts comprehensively. We first discuss the Temporal Dimension, then elaborate our considerations on Spatial Dimension, and finally summarize the design for the collective impacts from both temporal and spatial dimensions.

#### 3.2.1 TEMPORAL DIMENSION: INFLUENCE TIME

Different from prior works, our work consider the temporal dimension of Feint impacts by emulating them in a *Dynamic Short-Long-Term* manner. The rationale behind such a design choice is that: the purpose of *Feint* is to obtain strategic advantages against the opponent in temporal dimension, aiming to benefit following attacks. Hence, the *Dynamic Short-Long-Term* temporal impacts of

*Feint* shall be (1) the actions that follow *Feint* actions (e.g. actual attacks) in a short-term period of time should have strong correlation to *Feint* ; (2) the actions in the long-term periods explicitly or implicitly depend on the effect of the *Feint* and its following actions; and (3) for different *Feint* actions in different gaming scenarios, the threshold that divides short-term and long-term should be dynamically adjusted to enable sufficient flexibility in strategy making.

For *Dynamic Short-Long-Term*, we first set up a short-term planning threshold $st$ to select the follow-up actions, which are decided by *Feint* policy $\pi_i^{'}$ at $t_0$. Note that actions $\{a_{t_0+1}^i, ..., a_{t_0+st}^i\}$ are strongly related to the *Feint* action $a_{t_0}^i$. For the actions bounded by the short-term threshold, a set of large weights $\alpha = \{\alpha_{t_0}, ..., \alpha_{t_0+st}\}$ are used to calculate the reward:

$$Rew_{short-term}(\pi_i^{'}, t_0, st, \alpha) = \alpha_t \sum_{t=t_0}^{t=t_0+st} R^i(s_t, a_t^i, a_t^{-i}) \tag{1}$$

since these actions are expected to deliver higher reward (i.e. the purpose of *Feint* is to obtain strategic advantages) via the *Feint* action. We then consider long-term planning after the short-term planning threshold $st$: we use a set of discount factor $\beta = \{\beta_{t_0+st+1}, ..., \beta_T\}$ on the long-term average reward calculation (proposed by Kim et al. (2022)), to distinguish these reward from short-term rewards:

$$Rew_{long-term}(\pi_i^{'}, t_0, st, T, \beta) = \beta_t \frac{1}{T} \sum_{t=t_0+st+1}^{T} R^i(s_t, a_t^i, a_t^{-i}) \tag{2}$$

where $T$ denotes the end time of the game.

Finally, we put them together to formalize the *Short-Long-Term* reward calculation mechanism, when an agent $i$ plans to perform a *Feint* action at time $t_0$ with a short-term planning threshold $st$ and the end time of game $T$ as:

$$Rew_{temporal}(\pi_i^{'}, t_0, st, T, \alpha, \beta) = \lambda_{short} Rew_{short\ term}(t_0, st, \alpha) + \lambda_{long} Rew_{long\ term}(t_0, st, T, \beta) \tag{3}$$

where $\lambda_{short}$ and $\lambda_{long}$ are weights for dynamically balancing the weight of short-term and long-term rewards for different gaming scenarios. $\lambda_{short}$ and $\lambda_{long}$ are initially set as $0.67$ and $0.33$ and are adjusted to achieve better performance with the iterations of training.

### 3.2.2 SPATIAL DIMENSION: INFLUENCE RANGE

In a Multi-Player Game (i.e. usually more than two players), the strict one-to-one relationship between two agents is not realistic, since an agent can impact both the target agent and other agents. Therefore, the influences to all other agents shall maintain different levels Liu et al. (2021). Therefore, our work includes the spatial dimension of Feint impacts by fusing spatial distributions. The key idea of this design is to combine spatial distribution with the influence range during the game. More specifically, we incorporate Behavioral Diversity from Liu et al. (2021), to mathematically calculate and maximize the diversity gain of *Feint* actions in terms of the influence range.

We formalize the influence range of an action policy on $K$ agent based on $S \times A_i \times ... \times A_K$, which follows a distribution of multi-to-one relationships $T \rightarrow (\alpha_1 T_{(i,1)}, \alpha_2 T_{(i,2)}, ..., \alpha_K T_{(i,K)})$. The influence distribution can have different factors in different gaming scenarios. We demonstrate a set of commonly used factors in boxing games Won et al. (2021b) where agent $i$ plays against opponent $-i$ : $V = (A_i^k, A_{-i}^j, positions(i, -1), orientations(i, -i), linear\_velocities(i, -i), angular\_velocities(i, -i))$, in which the factors represent the chosen action $k$ of agent $i$, the chosen action $j$ of opponent $-i$, the relative positions, the relative moving orientations, the linear velocities and angular velocities of agent $i$ and opponent $-i$. When a *Feint* policy $\pi_i'$ is added, we aim to maximize the effective influence range under the influence distribution of *Feint* . Assuming the row agent $i$ maintains a policy pool $\mathbb{P}_i = \{\pi_i^1, \pi_i^M\}$, such influence distribution can be fused into Behavior Diversity measurement of the effective influence range by maximizing the discrepancy between the old influence effectiveness of policy occupancy measure $\rho_{\pi_E}(T)$ and the influence effectiveness when adding *Feint* policy of new policy occupancy $\rho_{\pi_i', \pi_{E-i}}(V')$:

$$max_{\pi_i'} Rew_{spatial}(\pi_i', V') = D_f(\rho_{\pi_i', \pi_{E_{-i}}}(V') \,||\, \rho_{\pi_E}(V)) \tag{4}$$

where the general $f$-divergence is use to measure the discrepancy of two distributions.

### 3.2.3 COLLECTIVE IMPACTS: INFLUENCE DEGREE

Solely relying on Temporal Dimension and Spatial Dimension overlooks the interactions between them, and these two dimensions are expected to have mutual influences for a realistic modeling Liu et al. (2021). Therefore, we consider the influence degree, so the collective impacts of these two dimensions can be aggregated in a proper manner.

We formulate the collective impacts for a *Feint* policy $\pi_i'$ in a Multi-Player Game that starts at $t_0$ and end at $T$ as:

$$Rew_{collective}(\pi_i') = \mu_1 \sum_{i=1}^{k} Rew_{temporal}(i, \pi_i', t_0, st, T, \alpha, \beta) + \mu_2 \sum_{t=t_0}^{st} max_{\pi_i'} Rew_{spatial}(\pi_i', V', t) \tag{5}$$

where temporal impacts $Rew_{temporal}$ (Section 3.2.1) are aggregated on spatial domain and spatial impacts $Rew_{spatial}$ (Section 3.2.2) are aggregated on temporal domain. $\mu_1$ and $\mu_2$ denote the weights of aggregated temporal impacts and spatial impacts respectively, enabling flexible adaption to different gaming scenarios. They are initially set as $0.5$ and are adjusted to achieve better performance with the iterations of training.

In addition to the collective impact of *Feint* itself in terms of temporal domain and spatial domain, our formalized *Feint* impacts can also result in response diversity of opponents, since different related opponents (spatial domain) at different time steps (temporal domain) can have diverse response. Such diversity can be used as a reward factor that make the final reward calculation more comprehensive Nieves et al. (2021); Liu et al. (2021). Thus, to incorporate such diversity together with our final reward calculation model, we refer to Liu et al. (2021) to characterize the diversity gain incurred by our collective impact formalization. When the impact $Rew_{collective}$ of *Feint* policy $\pi^{M+1}$ in a $M \times N$ payoff matrix $A_{\mathbb{P}_i \times \mathbb{P}_i}$ at when opponents choose policy $\pi_{-i}^{j}$ is collectively calculated, the derived diversity gain can be measured as follows:

$$Rew_{collective-diversity}(\pi_i^{M+1}) = D(a_{M+1} \,||\, A_{\mathbb{P}_i \times \mathbb{P}_i}) \tag{6}$$

$$a_{M+1}^{T} := (Rew_{collective}(\pi_i^{M+1}, \pi_{-i}^{j}))_{j=1}^{N}. \tag{7}$$

where $D(a_{M+1} \,||\, A_{\mathbb{P}_i \times \mathbb{P}_i})$ represents the diversity gain of the *Feint* action on current policy space. We follow the method in Liu et al. (2021) for the quantification of diversity gain.

## 4 FEINT IMPLEMENTATIONS

### 4.1 IMPLEMENTING THE FORMALIZATION AS A COLLECTIVE REWARD CALCULATION

To provide a comprehensive reward calculation model for *Feint* in Multi-Player Games, we synthesize the above collective impacts and collective diversity gain into the overall Reward Calculation Model. The robustness of similar design idea is proved in Liu et al. (2021) that the synthesised direct impacts and diversity gain can provide a more comprehensive reward calculation model for each player. Thus, we synthesize the collective impacts (Equation 5) and collective diversity gain (Equation 6) for a *Feint* policy $\pi_i'$ into the overall Collective Reward Calculation Model by applying weighted sum $\lambda_1$ of collective impact and $\lambda_2$ of collective diversity gain:

$$Rew^i(\pi_i') = \lambda_1 Rew_{collective}(\pi_i') + \lambda_2 Rew_{collective-diversity}(\pi_i') \tag{8}$$

### 4.2 FOR INFERENCE LEARNING MODELS

Inference Learning module is used to predict whether an observed action is *Feint* or not and policy parameters of $\theta^{-i}$ and policy dynamics $U^{-i}$. Model-based approaches use an explicit model to

fit an agent with the learning strategies of other agents from the observation Kim et al. (2021) but often suffer from the infinite recursion problem when an agent the model models the agent it self Tesauro (2003). Blei et al. (2016) proposes a model-free approach using approximate variational inference and Kim et al. (2022) optimizes a tractable evidence lower bound to infer accurate latent strategies of others. We add a random weight onto the ELBO to fit the randomness and uncertainty incurred by *Feint* actions. Specifically, the random-weighted ELBO is defined together with an encoder $p(\hat{z}_{k+1}^{-i} \mid \tau_{0:t}^i; \phi_{enc}^i; \gamma_{enc}^i)$ and a decoder $p(a_t^{-i} \mid s_t, \hat{z}_t^{-i}; \phi_{dec}^i; \gamma_{dec}^i)$ parameterized by a set of encoder and corresponding random weighted decision parameters $\{\phi_{enc}^i; \gamma_{enc}^i\}$ and a set of decoder and corresponding random weighted decision parameters $\{\phi_{dec}^i; \gamma_{dec}^i\}$:

$$
\begin{aligned}
J_{elbo}^i = \mathbb{E}_{p(\tau_{0:t}^i), p(\hat{z}_{0:t}^i \mid \tau_{0:t}^i; \phi_{enc}^i; \gamma_{enc}^i)} \Big[ \sum_{k=0}^{t-1} \log p(a_k^{-i} \mid s_k, \hat{z}_k^{-i}; \phi_{dec}^i; \gamma_{dec}^i) \\
- D_{KL}(p(\hat{z}_{k+1}^{-i} \mid \tau_{0:k}^i; \phi_{enc}^i; \gamma_{enc}^i) \parallel p(\hat{z}_k^{-i})) \Big]
\end{aligned}
\tag{9}
$$

where $\hat{z}_t^{-i}$ are latent strategies that represent inferred policy parameters of other agents $\theta_t^{-i}$ and $\tau_{0:t}^i = \{s_0, a_0^i, a_0^{-i}, r_o^i, ..., s_t\}$ denotes $i$'s trajectories up to timestep $t$. The random weight parameters $\gamma_{enc}^i$ and $\gamma_{dec}^i$ enable agents to randomly guess the probability of whether an action is a *Feint* action or not from the observations, since no inferred policy can properly select a *Feint* action.

## 4.3 FOR REINFORCEMENT LEARNING MODELS

Reinforcement Learning module updates the policy using various gradient ascending mechanisms Lowe et al. (2017); Iqbal & Sha (2019); Yu et al. (2021). Although these designs are different, one of the key components in policy updating is the reward function $Q$, which directly evaluates the reward of the policy and guides the policy updating process. Our proposed reward calculation mechanism (Section 3) can thus be fused into the MARL model by replacing the $Q$ functions. Such replacement can directly provide temporal, spatial and their collective considerations in policy making and keep the main structures of current MARL models. Thus, *Feint* actions can be effectively fused into policy learning process in current MARL models based on our reward calculation mechanism. We demonstrate the feasibility of fusion using two state-of-the-art MARL models, MADDPG Lowe et al. (2017) and MAAC Iqbal & Sha (2019) below.

In a $K$-agent game, agents $\{a_1, ..., a_k\}$ maintain policies $\pi = \{\pi_1, ..., \pi_k\}$ parameterized by $\theta = \{\theta_1, ..., \theta_k\}$, where the policies for other agents are inferred from the Inference Learning module 4.2. Each agent $i$ evaluates the expected reward $Q_i^\pi(s, a_1, ..., a_n)$ at state $s$ using reward term $Rew^i(s)$ in Equation 8.

In MADDPG Lowe et al. (2017) model, centralized critic $Q_i^\pi(s, a_1, ..., a_n)$ is used on decentralized execution to provide global information that can stabilize training. The centralized critic can be replaced by our reward calculation mechanism and the gradient descent of the policy learning is:

$$
\nabla_{\theta_i} J(\theta_i) = \mathbb{E}_{s \, p^u, a_i \, \pi_i} [\nabla_{\theta_i}] \log \pi_i(a_i \mid o_i) Rew^i(s)
\tag{10}
$$

where $Rew^i(s)$ is naturally fused into the gradient update function and provide influence range, influence degree and influence time length for adjusting learning outcomes.

In MAAC model Iqbal & Sha (2019), although the policy update mechanisms vary, the key part of policy updates is still the $Q_i^\pi(s, a_1, ..., a_n)$ function, which can also be replaced by our reward calculation term. The slight difference is that MAAC uses a temperature parameter $\alpha$ to balance between the entropy and rewards. Therefore, the gradient ascent for updating policy is:

$$
\nabla_{\theta_i} J(\theta_i) = \mathbb{E}_{s \, p^u, a_i \, \pi_i} [\nabla_{\theta_i}] \log \pi_i(a_i \mid o_i)(-\alpha \log \pi_i(a_i \mid o_i) + Rew^i(s))
\tag{11}
$$

where $Rew^i(s)$ replace the original $Q_i^\pi(s, a_1, ..., a_n)$ function and the multi-agent advantage function $b(o, a_1)$, since the expected return of multi-action interactions are already comprehensively calculated in $Rew^i(s)$. Thus, our proposed reward calculation can be seamlessly fused into state-of-the-art MARL models with simple replacement of the original $Q$-function while guaranteeing the feasibility.

## 5 EXPERIMENTAL METHODOLOGY

**Testbed Implementations.** We implement two complex Multi-player Games to examine the effectivenss of our design. We first re-implement and extend a strategic real-world game, AlphaStar Arulkumaran et al. (2019), which is widely used as the experimental testbed in recent studies of Reinforcement Learning studies Risi & Preuss (2020); Liu et al. (2021). We make extra efforts to emulate a six-player game, where players are free to have convoluted interactions with each other. And we implement *Feint* as dynamically generated policies, based on the 888 regular gaming policies. Then, we create a complex multi-player tagging game, based on Multi-Agent Particle Environment Mordatch & Abbeel (2017), an open-source environment from OpenAI. We handcraft a tagging game scenario, where six agents can freely fight with each other with 30 nuanced and flexible actions. Such an implementation requires an extensive amount of efforts since current available codebases only have a limited set of actions, which are insufficient to demonstrate the impacts of *Feint* . We follow the methodology from Wampler et al. (2010); Won et al. (2021a) to form *Feint* actions, based on 30 hand-crafted actions. Our tagging game resembles intense free fight scenarios in ancient Roman free fight scenarios Matz (2019), where interactions are intense and *Feint* is expected to be effective.

**Experiment Procedure.** We choose MADDPG Lowe et al. (2017) and MAAC Iqbal & Sha (2019) model in our experiments. We first train all six agents without *Feint* from our formalization on the state-of-the-art MARL models. Then we randomly select 3 agents (always labeled as Agent 1, 2, and 3), who incorporate our formalization of *Feint* , and keep the other 3 agents regular. All experiments are done with 4,000 training iterations on each model and 150 gaming iterations.

**Evaluation Metrics.** We examine the effects of *Feint* using ❶ gaming rewards of training, ❷ diversity gain of policy space and ❸ overhead of computation load. We first examine the learning outcomes (i.e. rewards) trained using both MADDPG and MAAC MARL model, by comparing the rewards of agents across all scenarios. We then examine the effects of *Feint* actions on how *Feint* can improve the diversity of gaming policies (Section 3). Finally, we perform overhead analysis, incurred by fusing *Feint* formalization in strategy learning.

## 6 EXPERIMENTAL RESULTS

### 6.1 GAMING REWARD GAIN

Figure 1 shows the rewards for each agent in two scenarios using MADDPG model. We make three observations. First, in ❶ in Figure 1, when no *Feint* is enabled, all agents' rewards tend to progress to a similar level when after enough training iterations. However, in ❶ in Figure 1, when *Feint* actions are enabled on agent 1, 2, and 3, these agents gain significantly higher rewards than the agents who are not enabled to perform *Feint* actions (agent 4, 5, and 6). Second, when comparing ❶ with ❷, the average rewards for agents who perform *Feint* actions (agent 1, 2, and 3 in ❷) is around 9.5, which is higher than the average rewards (around 5.0) for agents who do not perform *Feint* actions (all agents in ❶). These two observations demonstrate that our formalized *Feint* can provide effective improvement for agents' rewards. Third, the training results before 1000 iterations are not stable for both scenarios. This is mainly because the natural characteristics of MADDPG training process in which the training stables generally after 2000 iterations Lowe et al. (2017).

Figure 2 shows the rewards for each agent in two scenarios using MAAC model. When comparing to the results trained with MADDPG model, though the rewards differ in specific numbers, similar trends and observations are shown. First, when no *Feint* are enabled, all agents' rewards tend to progress to a similar level when after enough training training iterations (❶ in Figure 2), while when *Feint* actions are enabled on agent 1, 2, and 3, these agents gain significantly higher rewards than the agents who are not enabled to perform *Feint* actions (agent 4, 5, and 6) (❷ in Figure 2). Second, when comparing ❶ with ❷, the average rewards for agents who perform *Feint* actions (agent 1, 2, and 3 in ❷) is around 10.0, which is higher than the average rewards (around 3.5) for agents who do not perform *Feint* actions (all agents in ❶). And third, training in MAAC also suffers unstable results before the first 1000 iterations. With the comparison of the results for MADDPG and MAAC model, we have an addition observation that our formalized reward calculation for *Feint* actions and the MARL fusion can be effectively adapted on current state-of-the-art MARL models, providing promising feasibility and scalability for extension studies.

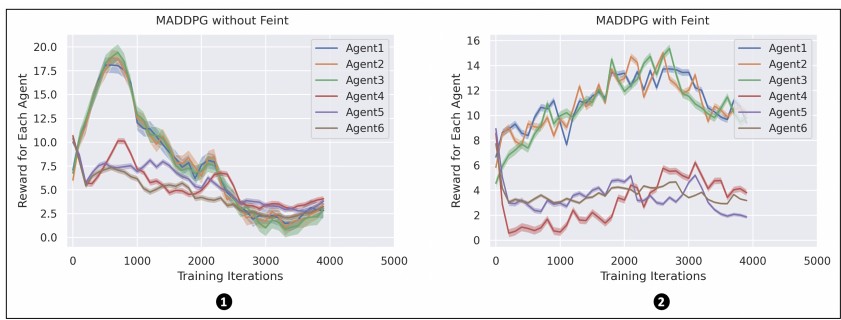

Figure 1: Reward for each agent in two scenarios trained by MADDPG. ❶ shows rewards for the first scenario where agents are not enabled *Feint* actions. ❷ shows rewards for the second scenario where agent 1, 2, and 3 are enabled *Feint* actions while agent 4, 5, and 6 are not enabled *Feint* actions.

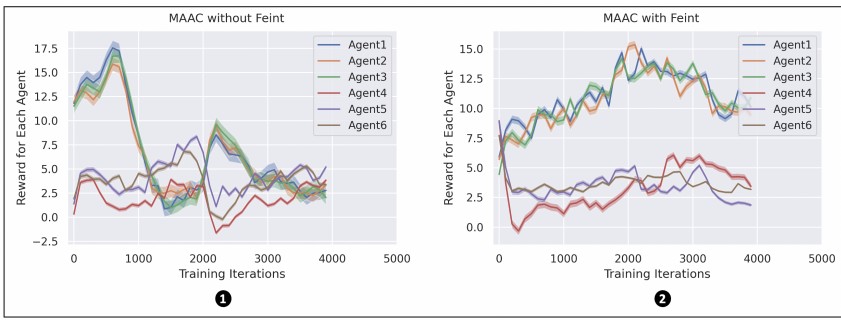

Figure 2: Reward for each agent in two scenarios trained by MAAC. ❶ shows rewards for the first scenario where agents are not enabled *Feint* actions. ❷ shows rewards for the second scenario where agent 1, 2, and 3 are enabled *Feint* actions while agent 4, 5, and 6 are not enabled *Feint* actions.

## 6.2 DIVERSITY GAIN

To examine the impacts on the policy diversity in games, we perform a comparative study between MARL training with and without *Feint* . Specifically, We use Exploitability and Population Efficacy (PE) to measure the diversity gain in the policy space. Exploitability Lanctot et al. (2017) measures the distance of a joint policy chosen by the multiple agents to the Nash equilibrium, indicating the gains of players compared to their best response. The mathematical expression of Exploitability is expressed as:

$$Expl(\pi) = \sum_{i=1}^{N}(max_{\pi_i'}Rew_i(\pi_i', \pi_{-i}) - Rew_i(\pi_i', \pi_{-i})) \qquad (12)$$

where $\pi_i$ stands for the policy of agent $i$ and $\pi_{-i}$ stands for the joint policy of other agents. $Rew_i$ denotes our formalized Reward Calculation Model (Section 4.1). Thus, small Exploitability values show that the joint policy is close to Nash Equilibrium, showing higher diversity. In addition, we also use Population Efficacy (PE) Liu et al. (2021) to measure the diversity of the whole policy space. PE is a generalized opponent-free concept of Exploitability by looking for the optimal aggregation in the worst cases, which is expressed as:

$$PE(\{\pi_i^k\}_{k=1}^N) = min_{\pi_{-i}}max_{1^\top\alpha=1 \ a_i>=0}\sum_{k=1}^{N}\alpha_k Rew_i(\pi_i^k, \pi_{-i}) \qquad (13)$$

where $\pi_i$ stands for the policy of agent $i$ and $\pi_{-i}$ stands for the joint policy of other agents. $\alpha$ denotes an optimal aggregation where agents owning the population optimizes towards. $Rew_i$ denotes our formalized Reward Calculation Model (Section 4.1) and opponents can search over the entire policy space. PE gives a more generalized measurement of diversity gain from the whole policy space.

Figure 3 shows the experimental results for evaluating diversity gains. From the figure, we obtain two observations. First, agents that can dynamically perform *Feint* actions (Agent 1, 2, and 3) achieve lower Exploitability (around $4.9 \times 10^{-2}$) compared to agents who perform regular actions (around $9.7 \times 10^{-2}$) and have higher PE (lower negative PE - around $5.3 \times 10^{-2}$) than those who only perform regular actions (around $1.2 \times 10^{-2}$). This result shows that our formalized *Feint* can effectively increase the the diversity and effectiveness of policy space. Second, agents with *Feint* have slightly higher variations in both metrics. This is because *Feint* naturally incurs more randomness (e.g. succeed or not) in games, resulting in higher variations in metrics.

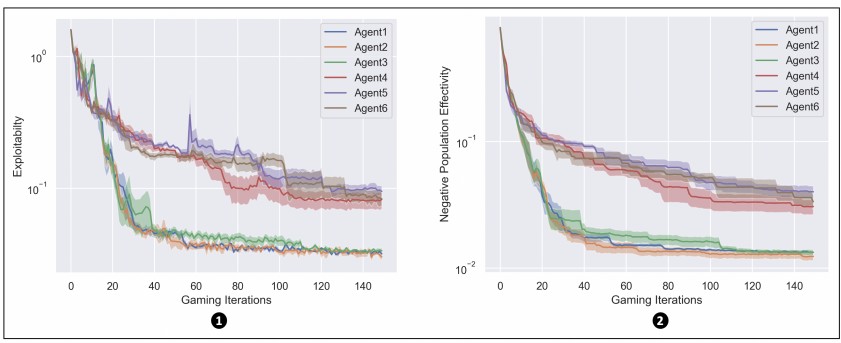

Figure 3: The difference for each agent, in terms of ❶ the exploitablity; and ❷ the negative population efficacy.

### 6.3 OVERHEAD ANALYSIS

Figure 4 shows the results of our overhead analysis. We make two observations. First, fusing *Feint* in MARL training do incur some overhead increment in terms of running time. This is because the formalization and fusion of *Feint* in MARL incur additional calculation load. Secondly, in both MADDPG models and MAAC models, the increased overhead is generally lower than $5\%$, which still indicates that our proposed formalization of *Feint* actions can have enough feasibility and scalability on fusing with MARL models.

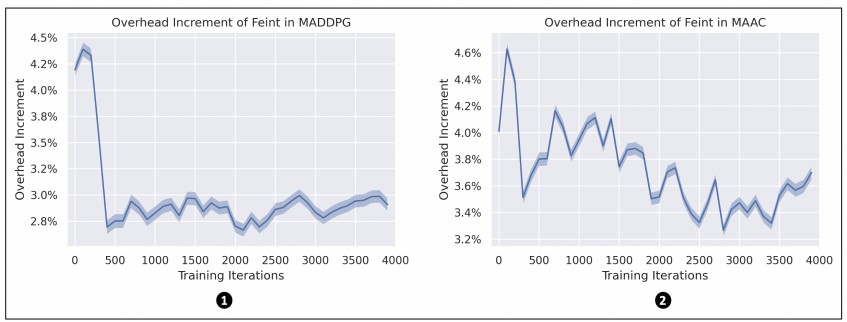

Figure 4: Overhead of *Feint* in ❶ MADDPG; and ❷ MAAC models.

### 7 CONCLUSION

We present the first formalization, implementation and quantitative evaluations of *Feint* in Multi-Player Games. Our work formalizes, implements and quantitatively examines *Feint* in Multi-Player Games, on the temporal, spatial and their collective impacts. The results show that our design of *Feint* can (1) greatly improve the reward gains from the game; (2) significantly improve the diversity of Multi-Player Games; and (3) only incur negligible overheads in terms of the time consumption. We conclude that our design of *Feint* is effective and practical, to make Multi-Player Games more interesting. Our design is also expected to be applicable for future models of Multi-Player Games.

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
