# OpenReview forum: "Feint in Multi-Player Games"
_ICLR.cc/2023/Conference — Submitted to ICLR 2023_

### Official Review · Reviewer_r9R4 · 2022-10-24

**Confidence:** 3
**Correctness:** 3
**Technical Novelty And Significance:** 2
**Empirical Novelty And Significance:** 3
**Recommendation:** 5

**Clarity, Quality, Novelty And Reproducibility:**

The paper is of good originality, however, some minor issues should be better explained to support the higher quality and clarity of the paper.

**Strength And Weaknesses:**

Strength:

- The proposed work’s motivation is explicit, the statement clearly proposed the problems to be solved and how to support the influence.
The authors pay effort into the multi-agent gaming’s feint’s formalization and tried to solve the problem in a more realistic scenario inspired by the previous two-agent feint work.

 - It is proven to be low timing cost while increasing the agent players’ diversity, encouraging it’s the possibility of deployment to other multi-agent tasks.


Weakness:

- For the results shown in Figure, are the second sub-figure guaranteed to be converged? It seems that the continuous tendency might result in a worse reward. Maybe the comparison of the results after a longer training iteration would be more convincing.

- The paper didn’t explicitly explain whether the two params “μ1 and μ2” are manually set weights or learned weights in Section 3.2.3. The same confusion also exists in part 4.1: ​​λ1 and λ2. If the parameters are empirical values, it’s better to be provided in the paper for better instruction on the method’s implementation.

- It’s stated in the experiments part that the environment Mordatch & Abbeel (2017), alphaStar, etc. were realized and tested to justify the Feint’s effectiveness. If the real scene of gaming can be provided in a case study or presentation video, the performance of diversity would be better verified.


**Summary Of The Paper:**

The proposed work intends to improve the diversity of multi-agent games by introducing a comprehensive Feint formalization in the combination of ‘temporal’, ‘spatial’, and ‘collective impact’; Despite some minor concerns about the experiments, the proposed method Feint’s effectiveness was proven through the overhead evaluation, which cost a relative low time-consumption to realize the diversity boosting and reward increase.

**Summary Of The Review:**

The work proposed an effective way to encourage the depiction of comprehensive feint, from spatial, temporal, and most innovative part, collective impact sides. According to the author’s explanation, the overall logic is clear and rational, however, as per the weakness, if the author could address the above concerns, the work will be more convincing.

---

> ### Author Response · Authors · 2022-11-10
> **Response to Reviewer r9R4**
>
> ## [Reviewer r9R4]
>
> ### - Are the second sub-figure in experiment results guaranteed to be converged?
> We assume so. The second sub-figures are guaranteed to converge at two levels: a higher gaming reward level for agents with Feint, and a lower gaming reward for agents without Feint.
>
> ### - What are the settings for parameter “μ1 and μ2” in Section 3.2.3 and “ λ1 and λ2” in Section 4.1?
> We point out that both “λ1 and λ2” in Section 4.1 and “μ1 and μ2” in 3.2.3 are dynamically changing for different Feint actions. The initial settings are set as 0.5 and adjust to achieve better performance with the iterations of training.
>
> ### - If the real scene of gaming can be provided in a case study or presentation video, the performance of diversity would be better verified.
> Our experiments focus on revealing the effectiveness of our design, by simulating gaming strategies in gaming scenarios with different reinforcement learning models. Since our work already requires extensive engineering efforts (as addressed in general comments), and the supports for graphics output hereby are not comprehensive.
>
> We agree with the reviewer on the need. We believe our work is essential to reveal the need for such supports, and the acceptance of our paper can attract more interested researchers and engineers to work on this together.

---

### Official Review · Reviewer_ebuk · 2022-10-24

**Confidence:** 2
**Correctness:** 4
**Technical Novelty And Significance:** 2
**Empirical Novelty And Significance:** 2
**Recommendation:** 5

**Clarity, Quality, Novelty And Reproducibility:**

Clarity: the ideas the paper presents lack better explanation.
Quality: the empirical section seems well presented.
Novelty: seems a bit limited; the idea is interesting yet the integration is rather trivial.
Replicability: the authors claim they hand-crafted the experimental domains, and the code does not seem to be available, which hinders replicability.

**Strength And Weaknesses:**

The idea of examining how bluffing in multi-player games may influence the game-play seems indeed compelling to me, yet I have to admit I find this work rather confusing. First, I believe the motivation could be greatly improved. Are the authors attempting to integrate feints, how they say, to “improve game experiences”? So do they imagine the applications of their work to be in the video-game industry, for example? Or is the inclusion of feints targeted more towards strategic robustness when multiple agents stray too far from the equilibrial paths, as examining reward gains in the experimental section seem to hint? The authors mention some “need for diversity” yet I do not entirely understand where this need comes from.

Second, I believe my confusions stems partly also from the language the authors are using. At times, I found myself going through some parts of the text multiple times because I felt like I understood the individual words, yet the sentences did not make much sense as they were worded together. One of the examples is “... [Naik et al.] … simply shrink smaller weights and deliver more future actions,” which I am not sure how to interpret. Overall, the text seem maybe a bit too ambiguous, and some more concrete or detailed explanation of, e.g., what temporal or spacial impact are could greatly help the readers to familiarize with these concepts and why the authors find them so important.

This initial exposition seems especially important in the light of the seemingly trivial integration of the authors’ concept into the feint detection or multiagent reinforcement-learning models.

I also lack some intuition why feints would help gaining more utility or achieve lower exploitability agains agents who are not using feints, as the experimental section seems to suggest. If I understanding the feints correctly, as explained in section 3.2.1, the bluffing agents sacrifice longterm full rationality for the possibility of obtaining short-term high rewards. Are the higher gains reported in the empirical evaluation the result of the agents not using feints being at this points still too far from some equilibrium? Or is it because there are multiple feinting agents in the environment, so the deviations are not unilateral?

Overall, I think the idea this paper presents is very interesting, but I feel the presentation needs to be further improved before I can support this paper's acceptance.

Nits:
Methods in 3.1 do not seem to relate to feint at all.
What are “Non-People Characters”? Do you mean “non-playable characters”?
“Multi-to-on”, maybe you mean “many-to-one”?
“we synthesis” to “we syntesize”


**Summary Of The Paper:**

If I understand this paper correctly, the authors introduce a way how to optimize for integrating “feint” actions into the solutions of stochastic-game-like interactions to promote diversity. They do so by gradually building up the criterion function from three aspects they consider the most important, and refer to them as “temporal”, “spacial”, and “collective” impacts. The first half of the paper is dedicated to formal mathematical formulation of each of these aspects. The authors then show how to integrate their build up reward function into reinforcement-learning algorithms, or feint recognition models. The last part of the paper presents empirical results achieved with agents using “feints”, examining gains in utilities, exploitability and diversity, and computational overheads incurred when optimizing for “feint” inclusion.

**Summary Of The Review:**

This paper presents a very interesting idea of including feints into agents’ strategies in multiplayer games, yet I feel the exposition need to be substantially improved for this paper to be considered at a conference like ICLR.

---

> ### Author Response · Authors · 2022-11-10
> **Response to Reviewer ebuk**
>
> ## [Reviewer ebuk]
> ### - Imprecise wording and typo: “Non-People Characters”, “Multi-to-on”, “we synthesis”
> To ensure the consistency of reading our response, we first clarify some imprecise wording. “Non-people Characters” should be “Non-Player Characters”; “we synthesis” should be “we synthesize”. However, we assume “Multi-to-one” is a proper word to state the relationship of multiple agents to one agent. We would revise them.
>
> ### - Why can Feint improve gaming experience and add diversity into games?
> Several works we mentioned in Section 2.2 highlight the importance of Feint in both real-world games and electronic games, since Feint can make the games less predictable and dominated. Current NPCs in electronic games still lack the level of diversity of humans in real-world games, since many intricate behaviors are not well-formalized. To this end, we view Feint as a representative tactical action that is ubiquitous and critical in many real-world games, yet are not well-formalized in game simulations.
>
> ### - Not understanding the sentences: “... [Naik et al.] … simply shrink smaller weights and deliver more future actions,”:
> By stating “shrink smaller weights and deliver more future actions”, we aim to explain that the weights for future actions’ rewards shrink exponentially with time in their work. We would revise it.
>
> ### - What temporal or spatial impact are could greatly help the readers to familiarize with these concepts and why the authors find them so important?
> For “temporal or spatial impact”, we highlight the necessity in Section 3.1 by addressing the limitations of current formalization of gaming actions. The temporal impacts is considered essential, because Feint itself cannot directly lead to reward gaining, which need to be combined with a set of following actions. The spatial impacts is considered essential in multi-agent scenarios because: when one agent perform a Feint on a targeted agent, other un-directly targeted agents may also be influenced and deviate their originally planned policies.
>
> ### - Why feints would help gaining more utility or achieve lower exploitability agains agents who are not using feints?
> Agents, who can not use Feint, are constantly deceived by other agents, who can use Feint. This has deviated policy from equilibriums. That's the rationale behind our evaluation results: the formalized Feint can effectively lead opponents to be deviated from equilibriums, and thus help the agents obtain strategic advantages.
>
> ### - Are methods in 3.1 relate to feint?
> Methods discussed in Section 3.1 are existing formalizations of normal actions and simple tactical actions, from which we justify that these formalizations are over-simplified and they can not be used for Feint formalization.

---

### Official Review · Reviewer_b6NC · 2022-10-26

**Confidence:** 3
**Correctness:** 1
**Technical Novelty And Significance:** 1
**Empirical Novelty And Significance:** 1
**Recommendation:** 1

**Clarity, Quality, Novelty And Reproducibility:**


It is hard to understand what the contribution of this paper is meant to be.  The main concept of "Feint" is not well defined, as far as I can tell. The originality of the work is very low, as it is a simple modification to the discounting of rewards at future states.

**Strength And Weaknesses:**

# Strengths
- The notion of feinting is interesting to consider in a multi-agent reinforcement learning environment.

# Weaknesses

- The notion of feint is never formally defined. So it is unclear exactly what behavior is sought from the agents, and how it differs from standard behavior. For example, if agents are learning optimal policies, wouldn't such policies include feinting if indeed it was beneficial?
- The proposed method to induce feinting is a simple modification of the reward function to emphasize near-term rewards over longer-term ones. Again, it is unclear how this is meant to encourage feints.
- The experimental results show that agents with the feinting behavior receive higher rewards. But, that is because their reward functions were directly modified to give different (larger values) to near-term rewards. So this is expected and an artifact of the definition -- it does not show that these agents actually have a competitive advantage, as far as I can tell.

**Summary Of The Paper:**

This paper proposes a method "Feint" to enable agents in multi-agent reinforcement learning to exhibit behaviors analagous to feints in human games; i.e. deceptive actions that are intended to look like an attack to draw a response from other agents.

**Summary Of The Review:**


Based upon the above comments concerning originality and clarity, I will recommend to reject the paper.

---

> ### Author Response · Authors · 2022-11-10
> **Response to Reviewer b6NC**
>
> ## [Reviewer b6NC]
>
> ### - This paper proposes a method "Feint" to enable agents in multi-agent reinforcement learning to exhibit behaviors analogous to feints in human games; i.e. deceptive actions that are intended to look like an attack to draw a response from other agents.
>
> We clarify that Feint is not a method. We refer the reviewer to the Cambridge Dictionary: Feint is an action “to pretend to move, or to make a move, in a particular direction in order to deceive an opponent, especially in sports such as football or boxing”. Our work is the first formalization of Feint, which can be incorporated with high-level learning strategies in Multi-Player Games.
>
>
> ### - What behavior is sought from the agents, and how it differs from standard behavior? For example, if agents are learning optimal policies, wouldn't such policies include feinting if indeed it was beneficial?
>
> We introduce Feint in Section 2.2 with examples (in sports games and electronic games). They are all active actions to obtain strategic advantages, thus having similar strategy-level formalization though with different concrete forms in different games. The learned policies are at strategy level, which are based on basic actions (e.g., swing sword and punching), in which Feint is neither included nor well-formalized. More importantly, we point out (in Section 3.1) that current policy learning can not support the full incorporation for the impacts of Feint.
>
> ### - Why Feint emphasize more on short-term rewards than long-term rewards?
> The purpose is to obtain the advantages by misleading the opponent, therefore the construction emphasizes short-term rewards.
>
>
> ### - Do the higher rewards for agents with Feint actions derived from manually giving different (larger values) to near-term rewards in reward functions?
> The rewards shown in the experiments are the actual gaming rewards (e.g., gain scores when hitting others and penalized by being hit), which is different from the reward function used to guide the policy making.

---

> > ### Comment · Reviewer_b6NC · 2022-11-21
> > **Response to rebuttal**
> >
> >
> > After reading the other reviews and the authors' rebuttal I still don't understand how increasing the short-term rewards incentivizes feinting. How does this cause an agent to pretend to make moves?
> >
> > My opinion remains unchanged and I will leave my score the same.

---

> > > ### Author Response · Authors · 2022-11-21
> > > **Clarifying the need of emphasizing short-term rewards**
> > >
> > > We clarify the need of emphasizing short-term rewards. The rationale of emphasizing short-term rewards is that: Feint actions cannot directly bring to gaming rewards. Following the definition of Feint, the actual gaming rewards come from the immediately-followed attack actions (which can be maximized by the previous Feint actions).
> > >
> > > Besides, we are not clear why the paper remains strong reject, and specific questions are welcome to follow up. We have clearly addressed the reviewer’s concerns about novelty, reward calculation mechanism, and experimental results in our rebuttal and revision, and there are no further questions on the above issues. Given the fact that the other two reviewers have exhibited clear understanding and appreciation of our novelty and proposed mechanism, we expect the reviewer to be specific on the concerns of our work and fairly examine our efforts.

---

### Author Response · Authors · 2022-11-10
**[Response to General Questions] Thanks for your comments.**

We thank all reviewers for valuable feedback. We are particularly excited to see Reviewer ebuk and r9R4 for appreciating the potentials of this work. We are continuing to revise the manuscript, and we would deliver a revision shortly (based on the feedback). Hereby, we address all concerns raised (and clarify some misunderstanding) from the reviewers.

## General Questions

### [Reviewer b6NC, ebuk] Novelty
We highlight that our work provides the first formalization, implementation, and evaluation of Feint. Reviewer b6NC misunderstands the concept of Feint, and we refer the reviewer to the Cambridge Dictionary: Feint is an action “to pretend to move, or to make a move, in a particular direction in order to deceive an opponent, especially in sports such as football or boxing”. The conceptual novelty of this work is to justify that Feint can be generalized to different reinforcement learning models, by accounting for the temporal, spatial, and the collective impacts.

Our extensive literature reviews justify the novelty. Prior works either (1) only focus on simplified two-agent scenarios (e.g. boxing and fencing), which overlook the spatial and collective impacts (as identified by our work) [1,2]; or (2) only consider game models in a simplified form, which can not account for the spatial and collective impacts [3,4].

Reviewers also argue that our work is a sole modification to the reward function, which is inaccurate and fair. We clarify that: though changes of the reward function seem straightforward, it has an extensive amount of implications on (1) the construction of multi-player gaming environments (since existing models are simplified, and we extensively engineer the Non-transitive Active Markov Game Model); (2) agent perceptions (since our formalization requires to account for the temporal and spatial impacts, which existing models cannot support it); (3) communication settings (since our formalization requires to account for the collective impacts, which existing models cannot support it). Though we highlight our conceptual contribution within the reward calculation, it does not mean that our work is a sole modification to it.

### References
[1] Won, J., Gopinath, D., & Hodgins, J. (2021). Control strategies for physically simulated characters performing two-player competitive sports. ACM Transactions on Graphics (TOG), 40(4), 1-11.

[2] Wampler, K., Andersen, E., Herbst, E., Lee, Y., & Popović, Z. (2010). Character animation in two-player adversarial games. ACM Transactions on Graphics (TOG), 29(3), 1-13.

[3] Lowe, R., Wu, Y. I., Tamar, A., Harb, J., Pieter Abbeel, O., & Mordatch, I. (2017). Multi-agent actor-critic for mixed cooperative-competitive environments. Advances in neural information processing systems, 30.

[4] Iqbal, S., & Sha, F. (2019, May). Actor-attention-critic for multi-agent reinforcement learning. In International conference on machine learning (pp. 2961-2970). PMLR.

---

### Decision · Program_Chairs · 2023-01-20

**Decision:**

Reject

**Justification For Why Not Higher Score:**

This paper seems clearly below bar.  Key concepts are not defined and the evaluation is not convincing.  All reviews were negative to one degree or another, including one very strong reject.

**Justification For Why Not Lower Score:**

n/a

**Metareview: Summary, Strengths And Weaknesses:**

(a) Summary: This paper aims to encourage agents to learn to "feint"; acting in a way that looks like a particular kind of action (e.g., attack).  Agents are encouraged to behave in this way by modifying short-term rewards.  Agents trained with feint modifications collected a larger  reward, although one reviewer points out that this is a questionable comparison given that the rewards have been modified.

(b) Strengths: All reviewers agreed that the idea of inducing "bluff"-like behavior is compelling, given that this is a widespread pattern in human behavior.

(c) Weaknesses: All reviewers had very serious clarity concerns.  A particular issue was that the notion of "feinting" was not formally defined in a sufficiently precise way; however, all reviewers were confused throughout the paper.


**Summary Of Ac-Reviewer Meeting:**

n/a